# Combined Neuro-Humoral Modulation and Outcomes in Patients with Chronic Heart Failure and Mildly Reduced or Preserved Ejection Fraction

**DOI:** 10.3390/jcm11226627

**Published:** 2022-11-08

**Authors:** Mauro Gori, Marco Marini, Lucio Gonzini, Samuela Carigi, Luisa De Gennaro, Piero Gentile, Giuseppe Leonardi, Francesco Orso, Denitza Tinti, Donata Lucci, Massimo Iacoviello, Alessandro Navazio, Enrico Ammirati, Annamaria Municinò, Manuela Benvenuto, Leonarda Cassaniti, Luigi Tavazzi, Aldo Pietro Maggioni, Renata De Maria

**Affiliations:** 1Heart Failure Working Group, Associazione Nazionale Medici Cardiologi Ospedalieri (ANMCO), 50121 Florence, Italy; 2Cardiology Division, Cardiovascular Department, Papa Giovanni XXIII Hospital, 24127 Bergamo, Italy; 3Department of Cardiovascular Sciences Cardiology, Ospedali Riuniti, 60122 Ancona, Italy; 4ANMCO Research Center, Heart Care Foundation, 50121 Florence, Italy; 5Cardiology Unit, Infermi Hospital, 47900 Rimini, Italy; 6Cardiology Department, San Paolo Hospital, 70123 Bari, Italy; 7De Gasperis Cardio ASST Grande Ospedale Metropolitano Niguarda, 20162 Milan, Italy; 8SSD Severe Heart Failure, PO “G. Rodolico”, 95125 Catania, Italy; 9Heart Failure Unit, Division of Geriatric Medicine and Intensive Care Unit, Department of Medicine and Geriatrics, Careggi University Hospital, 50141 Florence, Italy; 10Unit of Cardiology, San Camillo Hospital, 00152 Rome, Italy; 11Cardiology Unit, Department of Medical and Surgical Sciences, University Hospital Policlinico Riuniti, University of Foggia, 71122 Foggia, Italy; 12Cardiology Division, Arcispedale S. Maria Nuova, Azienda USL di Reggio Emilia—IRCCS, 71013 Reggio Emilia, Italy; 13De Gasperis Cardio Center, and Transplant Center, Niguarda Hospital, 20162 Milano, Italy; 14Department of Cardiology, A. Gallino Hospital, ASL3, 16164 Genoa, Italy; 15ICU Cardiology and Hemodynamics, G. Mazzini Hospital, 64100 Teramo, Italy; 16Cardiology Division, A.O. of National Importance and High Specialization “Garibaldi”, “Garibaldi-Nesima” Hospital, 95122 Catania, Italy; 17Maria Cecilia Hospital, GVM Care and Research, 48033 Cotignola, Italy

**Keywords:** neuro-humoral-modulating drugs, heart failure, mildly reduced ejection fraction, preserved ejection fraction, outcome

## Abstract

Pharmacotherapy of chronic heart failure with mildly reduced (HFmrEF) and preserved ejection fraction (HFpEF) remains challenging. We aimed to assess whether combined neuro-humoral modulation (NHM) (renin–angiotensin system inhibitors, betablockers, mineralocorticoid receptor antagonists) was differentially associated with outcome according to phenotype and age groups. Between 1999 and 2018 we recruited in a nationwide cardiology registry 4707 patients (HFmrEF *n* = 2298, HFpEF *n* = 2409) from three age groups: <65, 65–79 and 80+ years old. We analyzed clinical characteristics and 1 year all-cause mortality/cardiovascular hospitalization according to none/single, any double, or triple NHM. Prescription rates of no/single and triple NHM were 25.1% and 26.7% for HFmrEF; 36.5% and 17.9% for HFpEF patients, respectively. Older age was associated with higher prescription of no/single NHM in HFmrEF (p_trend_ = 0.001); the reverse was observed among HFpEF (p_trend_ = 0.005). Triple NHM increased over time in both phenotypes (all *p* for trend < 0.0001). Compared to no/single NHM, triple, but not double, NHM was associated with better outcomes in both HFmrEF (HR 0.700, 95%CI 0.505–0.969, *p* = 0.032) and HFpEF (HR 0.700, 95%CI 0.499–0.983, *p* = 0.039), with no interaction between NHM treatment and age groups (*p* = 0.58, *p* = 0.80, respectively). In a cardiology setting, among HF outpatients with EF > 40%, triple NHM treatment increased over time and was associated with better patient outcomes.

## 1. Introduction

Population ageing and the increasing prevalence of comorbidities concur with the projected increase in the absolute number of hospital admissions for patients with heart failure (HF) with mildly reduced (HFmrEF) and preserved ejection fraction (HFpEF) in the future [1].

Sodium–Glucose Cotransporter-2 Inhibitors (SGLT2-i) are the first drug class associated with improved outcomes in patients with left ventricular ejection fraction (LVEF) >40% [2]. Further to the documentation of efficacy in large randomized controlled trials, SGLT2i now have a class IIA guideline recommendation for the treatment of HFmrEF and HFpEF patients. Unfortunately, due to regulatory delays, it is not yet possible to prescribe these drugs for this indication in many countries, at least in patients not affected by diabetes mellitus. Conversely, drugs effecting neuro-humoral modulation (NHM), including RASI, BB, MRA, and angiotensin receptor neprilysin inhibitors (ARNI) have some suggestion of efficacy in decreasing cardiovascular hospitalization or mortality only in patients with HFmrEF, but not HFpEF, from post-hoc analyses of HF trials and have been included in the guidelines with a weaker class of evidence (IIB) [3,4,5,6,7,8]. Thus, nowadays there is still an unmet need to find effective pharmacological strategies in HFmrEF and HFpEF.

Contemporary HF guidelines for HFmrEF and HFpEF issue a class I recommendation for loop diuretics as clinically needed, together with physical training, appropriate lifestyle, nutritional changes, and the treatment of cardiac and non-cardiac comorbidities [7,8]. In this respect, renin–angiotensin system inhibitors (RASI), betablockers (BB), and mineralocorticoid receptor antagonists (MRA) are recommended for the treatment of cardiac conditions, including hypertension, chronic coronary syndrome, post-myocardial infarction, and rate control in atrial fibrillation.

Whether the combination of two or three agents realizing double or triple NHM might be effective in HFmrEF and/or HFpEF, overall and according to age, remains unclear.

We aimed to assess whether various NHM levels were differentially associated with all-cause mortality/cardiovascular hospitalization among HF patients with EF > 40% enrolled in a nationwide cardiology registry.

## 2. Materials and Methods

This report is a sub-analysis of patients with HFmrEF (LVEF 41–49%) and HFpEF (LVEF ≥ 50%), who were enrolled between 1999 and 2018 in the IN-HF registry, comprising a total of 150 centers geographically distributed over the whole country, as previously described [9,10], Appendix A.

The investigation conforms with the principles outlined in the Declaration of Helsinki. The protocol was approved by the Institutional Review Board of each participating center. All patients provided informed consent for scientific use of their clinical data collected in an anonymous way.

The diagnosis of HF was based on the presence of typical symptoms/signs with documented systolic and/or diastolic cardiac dysfunction [9,10]. Patients were enrolled upon their first presentation to the participating cardiology center, irrespective of disease stage and duration of HF history. Clinical management was based on the judgment of the attending physicians, who were not asked to do anything out of what they felt useful for the individual patient. For each patient, demographics, clinical history (including previous hospital admissions for HF), NYHA class and primary etiology of chronic HF, as previously defined [9,10], were entered in the database.

In the present analysis we have considered, as NHM, three classes of drugs, namely BB, RASI, and MRA. NHM is categorized as a treatment with none/a single one of the above agents, any combination of two NHM, or triple therapy. Non-NHM polypharmacy is defined as the daily intake of more than five drugs, excluding from the computation NHM.

Data on mortality and hospital admissions were obtained from hospital discharge ICD codes and primary-care physicians, complemented by information from patients and relatives when necessary. Although no provision was made for endpoint validation, specific training to uniform data collection was imparted at the beginning of the study [11]. The primary endpoint of our analysis was the combination of all-cause mortality or hospitalization for CV causes, which were coded as worsening chronic HF, ischemic events, arrhythmias, syncope, thromboembolism or stroke. Since HFpEF is a syndrome with a great burden of non-CV deaths, the combination of CV events (including HF) and all-cause death (including the non-CV part) could be considered as the most informative in this population.

The present analysis focuses on patients who had echocardiographic documentation of an LVEF > 40% at recruitment and prospective follow-up information at 1 year since enrolment.

Based on age at the time of recruitment, we classified patients into 3 categories (<65, 65–79, and ≥80 years old) according to the WHO definition [12]. Furthermore, to evaluate changes in prescription rates over epochs, we divided our population into 3 cohorts according to the time of recruitment (1999–2005, 2006–2011, and 2012–2018).

### Statistical Analysis

Categorical variables are reported as number and percentages, and compared by the Chi-squared test, whereas continuous variables were reported as means and standard deviations (SD), and compared by analysis of variance (ANOVA), if normally distributed, or by Kruskall–Wallis test, if not. There were no missing data for clinical variables. Trends for NHM therapy levels with respect to age categories and to study cohorts were tested by Cochran-Armitage test for binary variables.

The relation of NHM with the primary study end-point was separately assessed in the HFmrEF and HFpEF groups by Cox regression, adjusting the model for the following variables: age category, sex, ischemic etiology, HF history >6 months, body mass index, systolic blood pressure, heart rate, NYHA class, HF admissions in the previous year, history of hypertension, history of atrial fibrillation, diabetes, stroke/transient ischemic attack, furosemide, and non-NHM polypharmacy. Furthermore, the same multivariable analysis was performed, inserting into the model the interaction term between age category and number of NHM drugs.

Direct, adjusted Kaplan Meier curves for all-cause mortality or cardiovascular hospitalization according to NHM therapy were obtained by a stratified Cox analysis, adjusting the model for age categories; a simultaneous *p* value was also obtained to test the null hypothesis of no difference among the curves.

All tests were two-sided; a *p* value < 0.05 was considered statistically significant. All the analyses were performed with SAS system software, version 9.4 (SAS Institute Inc., Cary, NC, USA).

## 3. Results

### 3.1. NHM Prescription and Distribution

During the study period, we recruited 2298 patients with HFmrEF and 2409 subjects with HFpEF. NHM prescription rates were 25.1% for none/single, 48.2% for double, and 26.7% for triple NHM in the HFmrEF group (Table 1), while they were 36.5% for none/single, 45.6% for double, and 17.9% for triple NHM among HFpEF patients, respectively (Table 2). An older age was associated with a higher prescription of no/single NHM among HFmrEF patients (p_trend_ = 0.001); the reverse was observed among HFpEF patients (p_trend_ = 0.005). Prescription of triple NHM increased over time in both phenotypes (all *p* for trend < 0.0001).

### 3.2. Clinical Characteristics According to NHM

Clinical characteristics of HFmrEF patients according to NHM are presented in Table 1. When compared to subjects on no/single or double NHM, patients on triple NHM were younger and had more commonly a longer history of HF, recent HF admission, pre-existing hypertension, history of atrial fibrillation, furosemide prescription, and enrolment in recent epochs. Patients on triple NHM had a higher body mass index, and a lower systolic blood pressure, heart rate, and LVEF values than those receiving no/single or double NHM. Overall, the proportion of subjects who had at least one of the cardiac comorbidities for which an NHM agent may be indicated (history of hypertension; atrial fibrillation; ischemic etiology) was 80%. Specifically, this proportion was 79% in patients on no/single NHM, and 81% and 79% among those on double and triple NHM respectively (*p* = 0.29).

Clinical characteristics of HFpEF patients according to NHM are presented in Table 2. Again, patients on triple NHM had more commonly been enrolled in recent epochs, hospitalized in the previous year, and received more often furosemide. Moreover, they had more often pre-existing hypertension, a history of atrial fibrillation, an ischemic etiology, higher body mass index, lower systolic blood pressure, and lower LVEF values than those on no/single or double NHM. However, HFpEF patients on triple NHM were also older than both the other age groups. Overall, the proportion of subjects who had at least one of the cardiac comorbidities for which an NHM agent may be indicated was 81% and specifically 74% in patients on no/single NHM, and 85% and 87% among those on double and triple NHM, respectively (*p* < 0.0001).

### 3.3. NHM and Outcome

At one year follow-up, overall, 335 patients with HFmrEF (14.6%) and 356 with HFpEF (14.8%) met the primary end-point. Among patients with HFmrEF, death or CV hospitalization occurred in 102 (17.7%) on no/single NHM, 158 (14.2%) on double and 75 (12.2%) on triple therapy, respectively. Among patients with HFpEF, 139 (15.8%) on no/single NHM, 163 (14.8%) on double and 54 (12.6%) on triple therapy died or were hospitalized for CV causes.

Age-adjusted Kaplan–Meier curves on all-cause mortality or CV hospitalization according to NHM are presented in Figure 1 (HFmrEF) and Figure 2 (HFpEF). A significant difference among NHM groups was observed in HFmrEF patients (*p* = 0.041).

After multivariable adjustment, taking as reference no/single-NHM, triple NHM treatment was associated with a lower risk of the combined end point, both in the HFmrEF (HR 0.700 95% CI 0.505–0.969, *p* = 0.032) and in the HFpEF group (HR 0.700 95% CI 0.499–0.983, *p* = 0.039). Double therapy was not linked to patient outcome either in patients with HFmrEF (HR 0.889, 95%CI 0.683–1.156) or in those with HFpEF (HR 0.946, 95%CI 0.746–1.199). When the interaction term between NHM and age groups was entered in the model, no significant interaction was observed, either for HFmrEF (*p* = 0.58) or for HFpEF (*p* = 0.80).

## 4. Discussion

In a nationwide cardiology registry, a substantial proportion of patients with HFmrEF (26.7%) and HFpEF (17.9%) enrolled over two decades were prescribed triple NHM treatment, the cornerstone of guideline-recommended HFrEF treatment. Patients with HFmrEF and HFpEF who received triple NHM had similar characteristics: they were sicker, had a higher comorbidity burden and a longer history of HF, and had been enrolled in more recent epochs than subjects on no/single or double NHM. Both HFmrEF and HFpEF patients on triple NHM had better outcome, compared to those with none/single or double NHM.

### 4.1. NHM Prescription in HF with EF > 40%

Until recently, randomized controlled trials have failed to demonstrate significant benefits of NHM in both HFmrEF and HFpEF patients, and current European guidelines underline the lack of life-saving therapies in this setting [2,13,14,15]. The most accepted explanation for these frustrating results has been the heterogeneity of patients with “preserved” EF (EF > 40% or >45% according to enrolment criteria of HFpEF trials), mostly related to comorbidities frequently encountered in these patients [16,17,18,19].

However, treatment of comorbidities is a class I recommendation for HFpEF and HFmrEF in the latest European and US HF guidelines [2,3]. Most of our patients with HFmrEF (80%) and HFpEF (81%) had at least one condition for which NHM agents might be indicated. Additionally, in the HFpEF group, the proportion of patients who had a cardiac comorbidity requiring a NHM drug was significantly higher among patients on double, and particularly on triple, NHM than in those on no/single NHM.

Although we have no information on HF course before our patients were enrolled in the registry, we speculate that triple NHM prescription might have been a carry-on effect linked to previous evidence of LVEF values < 40% in the patient’s history, since temporal changes in LVEF are common among HF patients [20]. This might represent the most likely explanation for triple NHM prescription, especially in patients with HFmrEF [20,21]. Evidence of a high relapse rate after NHM withdrawal in HFrEF patients with improved LVEF (HFimpEF) likely underlies clinicians’ reluctance to stop these agents [22]. The lower LVEF values observed in both our HFmrEF and HFpEF patients on triple NHM with respect to those on double/single/no NHM might support this hypothesis.

Importantly, in both HFmrEF and HFpEF groups, triple NHM became more commonly prescribed in recent years (2012–2018), compared to previous epochs. This trend may reflect clinicians’ deep-rooted convincement that NHM could provide some benefit also in patients with EF > 40%, as supported by the baseline therapeutic findings in patients enrolled in DELIVER [23]. In fact, approximately 70–80% of patients in this contemporary HFpEF trial were on RASI, 60–70% were on BB, while 30% received MRA. Changing demographics over the last two decades is another possible explanation, with an overall longer life of HF patients that commonly entails more comorbidities, which may benefit from RASI, BB, or MRA [24].

Finally, post hoc analyses of randomized clinical trials (RCT), such as TOPCAT, PARAGON-HF, and CHARM Preserved [5,6,7,8,25] have suggested that NHM might be important, at least in HFmrEF, and current guidelines recommend the different NHM drugs with a IIB class of evidence in this phenotype [2,3]. Thus, trial results and expert opinions might have influenced increasing NHM prescriptions in the most recent epoch [8].

### 4.2. Factors Underlying Benefits of Combined NHM in HF with LVEF > 40%

One key finding of our study is the association of triple NHM with better outcomes in both HFmrEF and HFpEF. A possible explanation is the high prevalence (81%), in the triple NHM subgroup, of cardiac conditions for which a NHM agent is indicated, a cue to the prognostic importance of the strict management of comorbidities. Conversely, in the HFmrEF group, a higher prevalence of HFimpEF among patients on triple NHM might underlie the lower incidence of death and CV hospitalization [21], although we cannot provide proof of this statement from the history prior to enrolment.

Patients on triple NHM, as compared to no/single or double NHM, had lower heart rates and blood pressure levels, which is another possible explanation for better outcomes [26,27]. Patients with HFmrEF and HFpEF generally have higher blood pressure values than those with HFrEF, hence they are more likely to tolerate triple NHM.

Furthermore, patients on triple NHM presented markers of more severe disease, such as more common HF hospitalization in the previous year and higher furosemide use, which posed them at higher risk of events [28]. This patient subset might gain the most from a combined therapeutic regimen, both in HFmrEF and HFpEF. In fact, while differences in age-adjusted Kaplan–Meier curves on all-cause mortality or CV hospitalization according to NHM were significant only in HFmrEF patients, after multivariable adjustment, including markers of disease severity and complexity, triple NHM treatment was associated with a better outcome, both in HFmrEF and in HFpEF patients. Importantly, even if double NHM therapy was not as effective as triple NHM in improving patient outcomes, a clear trend of a possible beneficial effect was present even with double treatment at least in HFmrEF (HR 0.889, 95% CI 0.683–1.156). Thus, further studies with larger populations might provide some insights regarding the possible role of specific combinations of double NHM therapies in this setting.

Of note, the potential benefits from a combination of BB, RASI, and MRA among patients with LVEF above 40% is indirectly supported by a recent post-hoc comparison of multiple RCTs [29], which considered the theoretical benefit provided by the administration of a triple therapy comprising ARNI, MRA, and SGLT2i. However, ARNIs are not recommended for HFpEF treatment and EMA has not approved them even in patients with HFmrEF. Additionally, SGLT2i drugs have been effective in trials, considering patients already treated with BB and/or RASI (approximately 70–80% of patients), while 30% were on MRA [23]. Our findings add to this trial evidence, representing the real-world setting, which is often quite different from trial populations. Moreover, RCTs are normally designed to test the effectiveness of a specific drug, while they usually do not formally assess whether a combination of therapies might prove useful. This is an unmet need in HFpEF and HFmrEF, and registry data might indeed help to provide an answer to this important question.

Limitations of our analysis should be noted. Since dose levels for every NHM agent were not available for all patients, we did not test the achievement of target doses. Thus, we can only underscore the importance of receiving all the three drug classes, but whether this is relevant at high or low doses remains uncertain. However, a recent publication suggests that having on board a higher number of different compounds is even more important than receiving less drugs at target doses [30]. Although we adjusted the impact of NHM for multiple covariates, residual confounding remains an important issue in this type of study. Patients were enrolled in a cardiology outpatient setting; further data in other community settings are needed to confirm our findings. Finally, laboratory data were consistently collected only in the two most recent cohorts, while natriuretic peptides were not systematically available even in them. Thus, HF diagnosis was based on signs/symptoms and echocardiographic data.

In conclusion, in a cardiology outpatient setting, triple NHM therapy was associated with better patient outcomes in both HFmrEF and HFpEF when compared to no/single NHM. These data inform the potential benefits of using a combination of BB, RASI, and MRA among patients with LVEF above 40%.

## Figures and Tables

**Figure 1 jcm-11-06627-f001:**
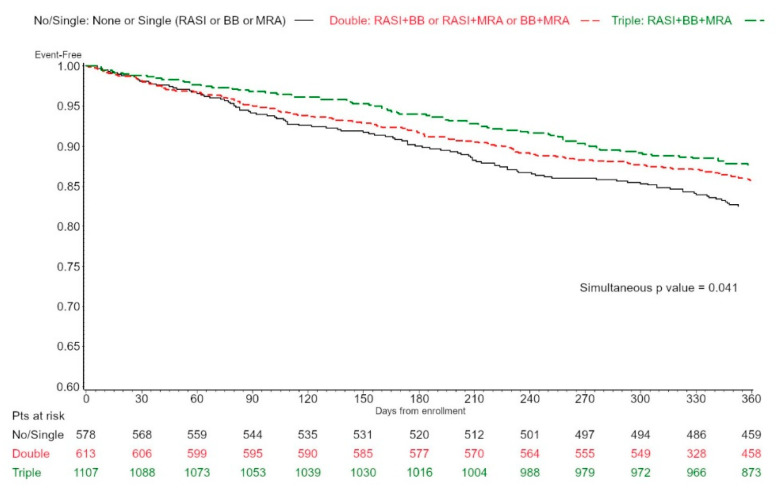
Age-adjusted Kaplan–Meier curves on all-cause mortality or cardiovascular hospitalization according to conventional NHM (RASI, betablockers, and MRA), none/single vs. double vs. triple, in 2298 patients with HFmrEF after adjustment by age categories. NHM, neurohumoral modulation.

**Figure 2 jcm-11-06627-f002:**
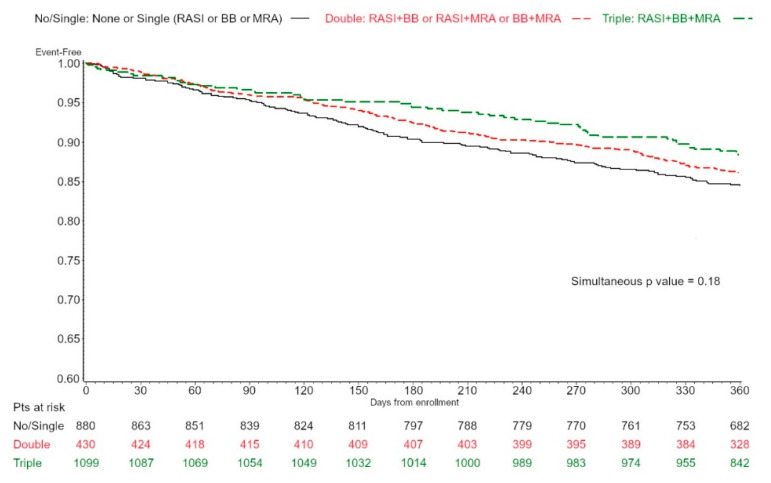
Age-adjusted Kaplan–Meier curves on all-cause mortality or cardiovascular hospitalization according to conventional NHM (RASI, betablockers, and MRA), none/single vs. double vs. triple, in 2409 patients with HFpEF after adjustment by age categories. NHM, neurohumoral modulation.

**Table 1 jcm-11-06627-t001:** Characteristics of HFmrEF patients according to conventional NHM.

LVEF 41–49%	All 2298	No-Single 578 (25.1)	Double 1107 (48.2)	Triple 613 (26.7)	*p* Value
Age (mean ± SD)	67 ± 14	68 ± 15	67 ± 13	67 ± 13	0.009
Age < 65	854 (37.2)	181 (31.3)	436 (39.4)	237 (38.7)	0.003
Age 65–79	1057 (46.0)	285 (49.3)	485 (43.8)	287 (46.8)	0.09
Age 80+	387 (16.8)	112 (19.4)	186 (16.8)	89 (14.5)	0.08
Female Sex	663 (28.9)	164 (28.4)	320 (28.9)	179 (29.2)	0.95
Epoch 1999–2005	782 (34.0)	299 (51.7)	381 (34.4)	102 (16.6)	<0.0001
Epoch 2006–2011	572 (24.9)	128 (22.1)	302 (27.3)	142 (23.2)	0.04
Epoch 2012–2018	944 (41.1)	151 (26.1)	424 (38.3)	369 (60.2)	<0.0001
HF admissions (previous year)	752 (32.7)	195 (33.7)	313 (28.3)	244 (39.8)	<0.0001
HF history >6 months	1396 (60.8)	314 (54.3)	670 (60.5)	412 (67.2)	<0.0001
History of hypertension	1349 (58.7)	309 (53.5)	667 (60.3)	373 (60.9)	0.01
Atrial fibrillation	665 (28.9)	166 (28.7)	293 (26.5)	206 (33.6)	0.007
Ischemic etiology	908 (39.5)	235 (40.7)	453 (40.9)	220 (35.9)	0.10
At least one cardiac comorbidity with an indication for NHM ^#^	1839 (80.0)	456 (78.9)	901 (81.4)	482 (78.6)	0.29
Diabetes	665 (28.9)	163 (28.2)	313 (28.3)	189 (30.8)	0.48
Previous stroke/TIA	163 (7.1)	46 (8.0)	82 (7.4)	35 (5.7)	0.27
Body mass index	27.2 ± 4.8	26.6 ± 4.7	27.1 ± 4.5	28.1 ± 5.3	<0.0001
LVEF%	44.6 ± 2.2	44.9 ± 2.3	44.6 ± 2.1	44.2 ± 2.1	<0.0001
Systolic blood pressure	130 ± 20	133 ± 21	130 ± 21	125 ± 18	<0.0001
Heart rate	70 ± 14	72 ± 15	70 ± 14	69 ± 13	<0.0001
NYHA class III–IV	307 (13.4)	103 (17.8)	126 (11.4)	78 (12.7)	0.001
Furosemide	1603 (69.8)	350 (60.6)	715 (64.6)	538 (87.8)	<0.0001
Non-NHM polypharmacy	418 (18.2)	101 (17.5)	187 (16.9)	130 (21.2)	0.07

Data are presented as *n* (%) or mean ± SD. NHM Neuro-Humoral Modulation. ^#^ history of HF/atrial fibrillation/ischemic etiology.

**Table 2 jcm-11-06627-t002:** Characteristics of HFpEF patients according to conventional NHM.

LVEF ≥ 50%	All 2409	No-Single 880 (36.5)	Double 1099 (45.6)	Triple 430 (17.9)	*p* Value
Age (mean ± SD)	70 ± 14	69 ± 16	71 ± 14	72 ± 13	0.007
Age < 65	701 (29.1)	281 (31.9)	309 (28.1)	111 (25.8)	0.04
Age 65–79	1017 (42.2)	372 (42.3)	458 (41.7)	187 (43.5)	0.81
Age 80+	691 (28.7)	227 (25.8)	332 (30.2)	132 (30.7)	0.06
Female Sex	1097 (45.5)	388 (44.1)	511 (46.5)	198 (46.1)	0.55
Epoch 1999–2005	668 (27.7)	347 (39.4)	279 (25.4)	42 (9.8)	<0.0001
Epoch 2006–2011	631 (26.2)	231 (26.3)	311 (28.3)	89 (20.7)	0.01
Epoch 2012–2018	1110 (46.1)	302 (34.3)	509 (46.3)	299 (69.5)	<0.0001
HF admissions previous year	803 (33.3)	261 (29.7)	366 (33.3)	176 (40.9)	0.0003
HF history >6 months	1589 (66.0)	562 (63.9)	727 (66.2)	300 (69.8)	0.10
History of hypertension	1504 (62.4)	467 (53.1)	730 (66.4)	307 (71.4)	<0.0001
Atrial fibrillation	986 (40.9)	337 (38.3)	449 (40.9)	200 (46.5)	0.02
Ischemic etiology	513 (21.3)	161 (18.3)	247 (22.5)	105 (24.4)	0.02
At least one cardiac comorbidity with an indication for NHM ^#^	1951 (81.0)	650 (73.9)	929 (84.5)	372 (86.5)	<0.0001
Diabetes	624 (25.9)	215 (24.4)	283 (25.8)	126 (29.3)	0.17
Previous stroke/TIA	182 (7.6)	70 (8.0)	87 (7.9)	25 (5.8)	0.32
Body mass index	27.1 ± 5.0	26.7 ± 4.9	27.2 ± 4.9	27.5 ± 5.3	0.01
LVEF%	57.1 ± 6.3	58.1 ± 6.7	56.8 ± 6.1	55.8 ± 5.6	<0.0001
Systolic blood pressure	130 ± 21	132 ± 21	130 ± 21	127 ± 19	0.001
Heart rate	72 ± 15	72 ± 15	71 ± 15	71 ± 14	0.24
NYHA class III–IV	473 (19.6)	184 (20.9)	217 (19.8)	72 (16.7)	0.20
Furosemide	1694 (70.3)	528 (60.0)	781 (71.1)	385 (89.5)	<0.0001
Non-NHM polypharmacy	470 (19.5)	161 (18.3)	214 (19.5)	95 (22.1)	0.27

Data are presented as *n* (%) or mean ± SD. NHM Neuro-Humoral Modulation. ^#^ history of HF/atrial fibrillation/ischemic etiology.

## Data Availability

The data underlying this article will be shared on reasonable request to the Board of the ANMCO Heart Failure Area.

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
