# Peer review of "Combined Neuro-Humoral Modulation and Outcomes in Patients with Chronic Heart Failure and Mildly Reduced or Preserved Ejection Fraction"

_jcm, 2022, doi:10.3390/jcm11226627_

Round 1

Reviewer 1 Report

The authors concentrate in the article on very crucial and important problem of HFmEF and HFpEF theraphy. Especially in HFpEF  to era of Sodium-Glucose Cotransporter-2 Inhibitors (SGLT2-i) there was no evidence on pharmacotherapy benefits. In this term study with large patients number gives interesting results. However  registry presents data from long period ago in which pharmacotherapy of HF definitevly changed, so the study does not reflect  present modern HFmEF and HFpEF therapy with SGLT2-i.  Anyhow authors present real life data, which are very important for future HF therapy discussion. Observation about low rate of optimal treatment in HF even in presence of indications as  comorbidities is crucial for organizing HF patients medical care. Benefit in  triple NHM group of  higher risk patients can be very important in creating next HF treatment standards. Minor revision of article should be done in points:

1.       Information in which percent of patients natriuretic peptides level was used   for heart failure diagnosis (what is present in last HF standards) should be provided

2.       the results presented  as well separately  as HF hospitalization and mortality would be very interesting

3.       no benefit of double NHM in comparision to triple NHM can be more extensively discussed 

4.        comparision of study results to RCTs data should be more widely discussed 

5.       subscription in 196 verse in the table 2- should be mentioned HFpef patients instead of patients

6.      

Author Response

The authors concentrate in the article on very crucial and important problem of HFmEF and HFpEF theraphy. Especially in HFpEF  to era of Sodium-Glucose Cotransporter-2 Inhibitors (SGLT2-i) there was no evidence on pharmacotherapy benefits. In this term study with large patients number gives interesting results. However  registry presents data from long period ago in which pharmacotherapy of HF definitevly changed, so the study does not reflect  present modern HFmEF and HFpEF therapy with SGLT2-i.  Anyhow authors present real life data, which are very important for future HF therapy discussion. Observation about low rate of optimal treatment in HF even in presence of indications as  comorbidities is crucial for organizing HF patients medical care. Benefit in  triple NHM group of  higher risk patients can be very important in creating next HF treatment standards. 

Response:

We thank the reviewer for the interest in our manuscript

Minor revision of article should be done in points:

  1. Information in which percent of patients natriuretic peptides level was usedfor heart failure diagnosis (what is present in last HF standards) should be provided

Response:

Unfortunately, only a minority of enrolled patients had data on BNP. Thus, as specified on page 2, the diagnosis of HF was based on the presence of typical symptoms/signs with documented systolic and/or diastolic cardiac dysfunction. The absence of BNP values for the diagnosis of HF has been quoted as a limitation in the track changes version, as follows:

Discussion (page 8, third paragraph): Finally, laboratory data were consistently collected only in the two most recent cohorts, while natriuretic peptides were not systematically available even in them. Thus, HF diagnosis was based on signs/symptoms and echocardiographic data.

  1. the results presented  as well separately  as HF hospitalization and mortality would be very interesting

Response:

This is another well taken point. For the present analysis, the primary endpoint was the combination of all-cause mortality or hospitalization for CV causes, which were coded as worsening chronic HF, ischemic events, arrhythmias, syncope, thromboembolism or stroke. This outcome is consistent with what has been already used in previous publications performed in this registry. Additionally, HFpEF is a syndrome with a great burden of non-CV death. Thus, the combination of CV events (including HF) and all-cause death (including the non-CV part) could be considered as the most informative in this population.

We have added the following to the Methods:

Methods (page 3, first paragraph): The primary endpoint of our analysis was the combination of all-cause mortality or hospitalization for CV causes, which were coded as worsening chronic HF, ischemic events, arrhythmias, syncope, thromboembolism or stroke. Since HFpEF is a syndrome with a great burden of non-CV death, the combination of CV events (including HF) and all-cause death (including the non-CV part) could be considered as the most informative in this population.

  1. no benefit of double NHM in comparision to triple NHM can be more extensively discussed

Response:

Thank you for this comment. We have added the following to our Discussion:

Discussion (page 8, first paragraph): In fact, while differences in age-adjusted Kaplan-Meier curves on all-cause mortality or CV hospitalization according to NHM were significant only in HFmrEF patients, after mul-tivariable adjustment, including markers of disease severity and complexity, triple NHM was associated to a better outcome, both in HFmrEF and in HFpEF patients. Importantly, even if double NHM therapy was not as effective as triple NHM in improving patient outcomes, a clear trend of a possible beneficial effect was present even with double treatment, at least in HFmrEF (HR 0.889, 95% CI 0.683-1.156). Thus, further studies with larger populations might provide some insights regarding the possible role of specific combinations of double NHM therapies in this setting.

  1. comparision of study results to RCTs data should be more widely discussed

Response:

We have added the following to the Discussion:

Discussion (page 8, second paragraph): Of note, the potential benefits from a combination of BB, RASI, and MRA among patients with LVEF above 40% is indirectly supported by a recent post-hoc comparison of multiple RCT, which considered the theoretical benefit provided by the administration of a triple therapy comprising ARNI, MRA, and SGLT2i. However, ARNIs are not recommended for HFpEF treatment and EMA has not approved them even in patients with HFmrEF. Additionally, SGLT2i have been effective in trials considering patients already treated with BB and/or RASI (approximately 70-80% of patients), while 30% were on MRA.

  1. subscription in 196 verse in the table 2- should be mentioned HFpef patients instead of patients

Response:

We have corrected this typo

Reviewer 2 Report

Dear authors

Thank you for allowing me to review this interesting retrospective study entitled ‘’  Combined neuro-humoral modulation and outcome in patients 2 with chronic heart failure and mildly reduced or preserved 3 ejection fraction’’. The study is focused on the effects of  neuro-humoral-modulation (renin-angiotensin system inhibitors, betablockers, mineralocorticoid receptor antagonists) on patients with HFmrEF and HFpEF. Study results conclude that over the decades, the percentage of patients on triple NHM blockage increased and that the triple therapy was associated with improved patient outcomes in the HFmrEF and HFpEF group of patients. The manuscript is well written and provides real world data on the role of neuro-humoral modulation on patients with HFmrEF and HFpEF. I have some minor comments:

1.      Abstract, line 26: Authors use the term ‘escalating’. I think this term causes confusion since it refers to the increase in the rates of usage of triple NHM medication over the decades instead of the intensification of NHM medication to the same patient. Authors perhaps need to make this more clear  throughout the abstract.  

2.      Main manuscript, Results, Line 196 : The term ‘’HFpEF’’ needs to be added following the word ‘’with’’ .

Author Response

Thank you for allowing me to review this interesting retrospective study entitled ‘’  Combined neuro-humoral modulation and outcome in patients 2 with chronic heart failure and mildly reduced or preserved 3 ejection fraction’’. The study is focused on the effects of  neuro-humoral-modulation (renin-angiotensin system inhibitors, betablockers, mineralocorticoid receptor antagonists) on patients with HFmrEF and HFpEF. Study results conclude that over the decades, the percentage of patients on triple NHM blockage increased and that the triple therapy was associated with improved patient outcomes in the HFmrEF and HFpEF group of patients. The manuscript is well written and provides real world data on the role of neuro-humoral modulation on patients with HFmrEF and HFpEF.

Response:

We thank the reviewer for the interest in our manuscript

I have some minor comments:

  1. Abstract, line 26: Authors use the term ‘escalating’. I think this term causes confusion since it refers to the increase in the rates of usage of triple NHM medication over the decades instead of the intensification of NHM medication to the same patient. Authors perhaps need to make this more clear  throughout the abstract.  

Response:

Thank you, we have made this clearer

  1. Main manuscript, Results, Line 196 : The term ‘’HFpEF’’ needs to be added following the word ‘’with’’ .

Response:

We have corrected this typo